# Modulation of In Vitro Macrophage Responses via Primary and Secondary Bile Acids in Dogs

**DOI:** 10.3390/ani13233714

**Published:** 2023-11-30

**Authors:** Alison C. Manchester, Lyndah Chow, William Wheat, Steven Dow

**Affiliations:** 1Department of Clinical Sciences, Colorado State University, Fort Collins, CO 80523, USAsteven.dow@colostate.edu (S.D.); 2Department of Microbiology, Immunology & Pathology, Colorado State University, Fort Collins, CO 80523, USA

**Keywords:** cytokine, macrophage, bile acid, cholic acid, lithocholic acid, receptor, transcriptome, dog

## Abstract

**Simple Summary:**

Bile acids (BAs) are compounds made by the liver that act within the intestinal lumen to aid fat digestion. These molecules are also important signals for the intestinal immune system. Primary BAs (e.g., cholic acid) are converted via intestinal bacteria to secondary BAs (e.g., lithocholic acid). The balance between these two classes of BAs is disrupted in dogs with chronic enteropathy, but the impact on gut immunity is unknown. Changes in diet and antibiotic treatment also disrupt gut luminal BAs by altering gut bacterial populations. Primary and secondary BAs are known from studies in other species to exert different effects on innate immune responses, but their role in canine immunity has not been explored. Therefore, we conducted studies to elucidate the effects of primary and secondary BAs on macrophage immune responses in dogs, with the goal of exploring their possible roles in intestinal immunity. We found some shared and some divergent effects of primary versus secondary BAs on canine macrophages. Our findings suggest that the secondary BAs play the dominant role in regulating GI inflammation in dogs.

**Abstract:**

Bile acids (BA) are important metabolites secreted into the intestinal lumen and impacted by luminal microbes and dietary intake. Prior studies in humans and rodents have shown that BAs are immunologically active and that primary and secondary BAs have distinct immune properties. Therefore, the composition of the gut BA pool may influence GI inflammatory responses. The current study investigated the relative immune modulatory properties of primary (cholic acid, CA) and secondary BAs (lithocholic acid, LCA) by assessing their effects on canine macrophage cytokine secretion and BA receptor (TGR5) expression. In addition, RNA sequencing was used to further interrogate how CA and LCA differentially modulated macrophage responses to LPS (lipopolysaccharide). We found that exposure to either CA or LCA influenced LPS-induced cytokine production via macrophages similarly, with suppression of TNF-α secretion and enhancement of IL-10 secretion. Neither BA altered the expression of the BA receptor TGR5. Transcriptomic analysis revealed that CA activated inflammatory signaling pathways in macrophages involving type II interferon signaling and the aryl hydrocarbon receptor, whereas LCA activated pathways related to nitric oxide signaling and cell cycle regulation. Thus, we concluded that both primary and secondary BAs are active modulators of macrophage responses in dogs, with differential and shared effects evident with sequencing analysis.

## 1. Introduction

Chronic enteropathy (CE) in dogs is thought to be driven by chronic intestinal inflammation, yet the pathogenesis of the disease is incompletely understood. As potential regulators of gut immunity, intestinal bile acids (BAs) are increasingly recognized to be important signaling molecules within and outside of the gut [1]. Bile acids mediate their activity in part through the G-protein-coupled receptor TGR5, which is expressed by both macrophages and intestinal epithelial cells [2]. In mice and humans, TGR5 activation plays a role in BA synthesis, energy homeostasis, hepatic regeneration, and inflammatory responses [3,4,5]. The secondary BA lithocholic acid (LCA) is the strongest endogenous agonist of TGR5 [6], with lesser agonist activity exerted by the primary BAs cholic acid (CA) and chenodeoxycholic acid (CDCA). Studies in rat, rabbit, and human cells have highlighted the immunomodulatory role of BAs, with suppression of LPS-stimulated (lipopolysaccharide) TNF-α production via macrophages exposed to LCA and CDCA but not CA [6,7,8,9].

Specific members of the intestinal microbiota perform critical BA transformation reactions, including 7α-dehydroxylation, which is necessary for primary BA conversion to secondary BAs [10]. Secondary BAs, including LCA, are the most abundant BAs in the distal gut lumen in health [11], but the balance of primary and secondary BAs may be disrupted in chronic intestinal diseases in humans (e.g., inflammatory bowel diseases, irritable bowel syndrome [12]) and in dogs [13,14]. These disruptions in BA pool makeup (specifically, expansion of primary BA relative to secondary BA) are associated with more severe intestinal inflammation [12,15,16] and dysbiotic excursions [13,17], which may play a role in the pathogenesis of CE in dogs. While the overall impacts of these changes in BA levels are incompletely understood, evidence from a recent study found that clinical improvement in dogs with CE fed a hydrolyzed protein diet was associated with increased concentrations of secondary BAs [13]. These findings suggest that restoring normal intestinal luminal BA composition may be an important mechanism by which therapeutic diets induce clinical remission from CE.

As an example of the relevance of intestinal BA profiles, we and others reported that oral administration of certain antimicrobials commonly used to treat dogs with CE, including tylosin and metronidazole, has been associated with changes in intestinal BA pool, with a shift towards a BA profile dominated by primary BAs [18,19]. Dogs with CE treated with antibiotics typically have poor long-term responses and struggle with recurrent GI signs [20,21]. The fecal microbiota of dogs with CE is characterized by reduced relative abundances of Firmicutes and Clostridia and increases in the abundance of Proteobacteria [22,23,24]. These alterations in bacterial relative abundances may be important as a diminished prevalence of anaerobic bacteria such as *Clostridium* and *Bacteroides* spp. able to perform BA biotransformation reactions [25,26] is correlated with a reduction in proportions of fecal secondary BAs [14]. Restoring the normal distal gut BA profile may therefore be an important therapeutic endpoint.

In the study reported here, we investigated and compared differences in innate immune responses to primary and secondary BAs in dogs. The studies used a canine macrophage cell line and primary cultures of monocyte-derived macrophages to determine the impact of BAs on LPS-induced macrophage cytokine production, along with transcriptomic responses. The study revealed distinct differences in innate immune responses to primary versus secondary BAs in dogs and suggested that these two classes of BAs may be important factors in regulating intestinal immunity in health and disease.

## 2. Materials and Methods

### 2.1. Biochemical Reagents

Bile acids (cholic acid (CA; C1129-25G); lithocholic acid (LCA; L6250-10G); sodium taurolithocholate (T-LCA; T7515-100MG); and chenodeoxcycholic acid (CDCA; C9377)) and sodium azide (S2002) were obtained from Sigma-Aldrich Co., St. Louis, MO, USA. Lipopolysaccharide from *E. coli* 055:B5 was sourced from InvivoGen, San Diego, CA, USA. Human macrophage colony stimulating factor (M-CSF; cat#300-25-50UG) was obtained from PeproTech, Thermo Fisher Scientific, Waltham, MA, USA.

### 2.2. Cell Culture Medium

Cells were cultured in Dulbecco’s modification of eagle’s medium with 4.5 g/L glucose, L-glutamine, and sodium pyruvate (DMEM; Corning, Manassas, VA, USA) with the addition of 10% fetal bovine serum (FBS; Peak Serum, Wellington, CO, USA), CTM, and 2-mercaptoethanol (Gibco by Life Technologies Corporation, Grand Island, NY, USA) at 55 mM. Cells were expanded in 300 cm^3^ vented-cap tissue culture flasks (Fisherbrand, Waltham, MA, USA) and plated in flat-bottom low-evaporation lid 24-well cell culture plates (Falcon, Corning Incorporated, Corning, NY, USA).

### 2.3. Canine Macrophage Cell Lines

Two canine macrophage tumor cell lines were used in these studies. One canine macrophage cell line (MH588, generated in the Dow lab) was isolated from a lymph node aspirate of a dog with malignant histiocytosis. The cell line was phenotyped using flow cytometry to confirm macrophage-like cell identity (expression of CD14 and CD11b; Appendix A). A second canine macrophage cell line (DH82) was obtained from the American Type Tissue Collection (Gaithersburg, MD, USA) and has previously been extensively characterized and used in canine macrophage assays [27,28,29]. The MH588 and DH82 cells were cultured in complete medium (see above) using incubators at 37 °C with 10% CO_2_. Cells were detached from flasks with trypsin for 10 min, resuspended in fresh medium, and seeded at 1 × 10^5^ cells/well in 24 well plates for cytokine release and RNA sequencing assays. Cells were cultured to 80% confluency prior to the initiation of treatment.

### 2.4. Generation of Monocyte-Derived Macrophages

Monocyte-derived macrophages (MDM) were generated, as previously reported [30]. Briefly, peripheral blood mononuclear cells (PBMC) were prepared from EDTA-collected blood samples from healthy dogs (CSU IACUC #1440) and seeded at a density of 2–5 × 10^6^ cells per ml in 48-well plates. After allowing cells to adhere for 4 h, the medium with non-adherent cells was gently removed and replaced with fresh DMEM containing M-CSF at 50 ng/mL. Cells were differentiated for 7 days, with media changes with M-CSF after 3 and 5 days. On day 7, the media was replaced, and cells were treated as noted.

### 2.5. Bile Acid Treatment of Macrophage Cultures

Triplicate wells of macrophages were treated with BAs at 25 μM; this concentration was selected to fall between estimates of BA concentrations in the intestinal lumen (<1000 to 10,000 μM), plasma BA concentrations (25 to 50 μM), and fecal BA concentrations in people (up to 5 mM) [31,32]. After 2 h of incubation, cells were activated with lipopolysaccharide (LPS) at 100 ng/mL; this concentration was selected as a conservative extrapolation from studies of bacterial-derived LPS levels in the gut lumen and within inflamed gut mucosa [33,34,35]. After 48 h of bile acid and LPS co-culture, supernatants were harvested and frozen at −20 °C until ELISA evaluation. The macrophage activation and cytokine release assays were repeated 6 times to ensure reproducibility. After supernatants were removed, cells were rinsed with PBS, then trypsinized for 10 min, and then resuspended in flow cytometry staining buffer (FACS) for flow cytometry (see below).

### 2.6. Flow Cytometry for Assessment of Macrophage Expression of the Bile Acid Receptor TGR5

Cells were collected following 48 h of stimulation with LPS ± BAs, and expression of the bile acid receptor TGR5 was measured using flow cytometry. The antibody used was a polyclonal IgG rabbit anti-human TGR5 antibody (Abcam, Cambridge, UK). The secondary antibody used was donkey anti-rabbit polyclonal IgG Cy3-conjugated (Jackson ImmunoResearch Laboratories, West Grove, PA, USA). Negative controls included cells labeled with isotype-matched antibodies, including ChromePure Rabbit IgG (Jackson ImmunoResearch Laboratories, West Grove, PA, USA).

To assess expression of TGR5, cells were incubated with anti-TGR5 antibody (diluted 1:100) together with 5% normal donkey serum (as a block for non-specific binding) for 30 min at 4 °C. Cells were then washed, incubated with secondary antibody (donkey anti-rabbit IgG), washed, and then resuspended in PBS with FACS buffer (PBS with 2% FBS and 0.1% sodium azide) for analysis. To gate out dead cells, 2% 7-aminoactinomycin D (7-AAD; Thermo Fisher Scientific, Waltham, MA, USA) was added just before analysis.

Immunostained cells were analyzed using a Beckman Coulter Gallios flow cytometer (Brea, CA, USA). The analysis was performed using FlowJo software version 10.10 (Ashland, OR, USA). The analysis included the determination of the percentage of positive fluorescent cells in addition to the geometric mean fluorescence intensity of labeled cells. Background fluorescence was determined using either unstained cells or cells stained with isotype-matched antibodies.

### 2.7. ELISA Assays for Canine Cytokines

Cytokine concentrations were measured using canine-specific ELISA assays (DuoSet ELISA canine IL-10 [DY735] and TNF-α [DY1507], R&D Systems, Minneapolis, MN, USA) according to the manufacturer’s protocol. Validation has been performed by the company, and this ELISA has been used extensively by our group and others [36]. Cytokine concentrations were extrapolated from a standard curve generated with recombinant cytokines as part of the ELISA kit. Absorbance was measured at 450 nm using a Biotek plate reader.

### 2.8. Assessment of TGR5 Expression via Fluorescence Microscopy

Macrophages were cultured on cell culture chamber slides (In Vitro Scientific, Sunnyvale, CA, USA) overnight, then fixed with 2% paraformaldehyde and immunostained with anti-TGR5 antibody resuspended in FACS buffer with 0.25% saponin for cell permeability, as described previously [30].

### 2.9. MTT Assay

Macrophages were cultured in a 96-well plate as described above prior to the addition of 3-(4,5-dimethylthiazol-2-yl)-2,5-diphenyltetrazolium bromide reagent (MTT) at 0.25 mg/mL to test for cell viability by assessing redox potential [37]. After 4 h at 37 °C, 100 μL of MTT stop (HCl) was added, wells were triturated vigorously, and absorbance was read using a plate reader at 570 nm.

### 2.10. RNA Sequencing and Analysis Pipeline

The impact of BA treatment on macrophage transcriptomic responses was assessed using RNA sequencing as described by us previously [30,38,39]. Briefly, MH588 cells were pre-incubated with BA for 2 h prior to stimulation with LPS and then cultured for another 24 h, at which point RNA was extracted from cells using a Qiagen RNeasy mini kit (Qiagen). Extracted RNA was sent to Novogene Corp. (San Diego, CA, USA) for sequencing using an Illumina platform. RNA quality was determined using the Agilent 4200 Bioanalyzer system. The sequence library was generated using the NEBNext Ultra II RNA Library Prep Kit (New England Biolabs, Ipswich, MA, USA) following the manufacturer’s instructions. Libraries were sequenced on an Illumina NovaSeq 6000 (Illumina, San Diego, CA, USA). Paired-end reads 150 bp in length were generated, and files were delivered as de-multiplexed fasq files. Analysis of RNA sequencing data was completed using Partek Flow software, version 10.0 (Partek Inc., Chesterfield, MO, USA). Raw reads were filtered for adapters and reads containing N > 10% and for Phred scores >30. Alignment was conducted using STAR 2.7.3a [40], and with the CanFam3.1 genome assembly, reads were annotated and counted using HT-seq with Ensembl 107 [41].

### 2.11. Statistical Analysis

Basic statistical analysis was completed in GraphPad Prism for macOS (version 10.0.1). Differences between cytokine concentrations with different treatments were assessed with a one-way ANOVA with *p* values adjusted for multiple comparisons. Adjusted *p* values < 0.05 were considered statistically significant. Expression of the BA receptor TGR5 was assessed qualitatively via immunocytochemistry (ICC) and quantitatively via flow cytometry; the numbers of positive cells were compared using a one-way ANOVA. For RNAseq analysis, GSA (gene set analysis) (https://documentation.partek.com/display/FLOWDOC/GSA accessed on 14 April 2023) was performed on normalized transcript counts, and differentially expressed genes were filtered using *p* values and fold change −1.5< or >1.5. The STRING database was used to assess inferred protein–protein interactions [42].

## 3. Results

### 3.1. Impact of Primary and Secondary BAs on Cytokine Secretion via LPS-Activated Canine MH588 Macrophages

The first question we addressed was how primary and secondary BAs may differentially impact cytokine secretion via TLR4-activated macrophages as a model for the modulation of immune responses via gut macrophages responding to TLR-activating stimuli such as LPS. To address this question, we used an in vitro model system (Figure 1) employing canine macrophages activated with LPS and co-incubated with two different primary BAs (CA and CDCA) and a free or taurine-conjugated secondary BA, LCA. For most of these studies, macrophages were pre-incubated with BAs for 2 h prior to LPS activation. However, in some experiments, the BAs were added simultaneously. We found the order of incubation did not affect overall cytokine responses. Previous studies with human macrophages have demonstrated that taurine-conjugated LCA suppresses pro-inflammatory cytokine production and gene expression via LPS-activated macrophages, supporting the generation of a more regulatory or immune-suppressive phenotype [9,43].

We observed first that incubation with primary or secondary BAs alone did not activate MH588 macrophages; specifically, neither release of TNF-α nor IL-10 was measurable in culture supernatants after 48 h of incubation with any of the 4 BAs screened. Therefore, LPS-activated macrophages were used for the rest of the studies reported here because LPS is a key immune-activating molecule to which macrophages in the GI tract are continuously exposed. When BAs were added to LPS-activated MH588 macrophages, cytokine production was modulated differently by primary vs. secondary BAs. For these studies, we measured a key pro-inflammatory cytokine (TNF-α) previously shown to be upregulated in intestinal biopsies of dogs and humans with enterocolitis [44,45]. Macrophage production of a key immune modulatory/suppressive cytokine (IL-10) was also measured to assess the potential immune suppressive properties of BAs.

We then determined whether there were concentration-dependent effects of BAs on LPS-stimulated TNF-α release via macrophages (Appendix A), based on estimated physiological concentrations of BAs in the intestinal lumen [32] and dose titration experiments in our canine macrophage system. In addition, we assessed the impact of BAs on macrophage viability using an MTT assay and found no impact at the concentrations tested, including 25 μM. Therefore, we selected 25 μM as the concentration that was used in the studies reported here. In initial screening experiments with three unconjugated BAs, we found that the most marked differences on cytokine release were observed between CA and LCA (Appendix A) following LPS activation, so these BAs were selected for subsequent studies. DH82 cells failed to produce either TNF-α or IL-10 following LPS stimulation, whereas MH588 cells produced strong TNF-α and IL-10 responses. Therefore, MH588 cells were used for the remainder of the studies shown here.

We first observed that incubation of LPS-activated MH588 macrophages with the primary BA CA did not alter macrophage secretion of TNF-α compared to LPS alone. The other primary BA, CDCA, induced a mild reduction in TNF-α release. We next examined the impact of LCA on macrophage cytokine responses following LPS activation. We found that treatment with the secondary BA LCA significantly suppressed TNF-α production and also increased the anti-inflammatory cytokine IL-10 release (Figure 2). Taurine-conjugated LCA (TLCA) suppressed TNF-α release but failed to stimulate IL-10 release (Figure 2b). These findings indicated that, like other species, the secondary BA LCA was immunologically active and significantly suppressed LPS-stimulated macrophage inflammatory responses. The primary BAs varied in their immunosuppressive qualities, with CDCA acting to suppress TNF-α release but neither CDCA nor CA modulating IL-10 release.

### 3.2. Impact of Primary and Secondary BAs on Cytokine Secretion via LPS-Activated Primary Monocyte-Derived Macrophages

We next assessed the impact of BAs on cytokine secretion by primary cultures of canine macrophages to ascertain whether they responded differently to BAs than the macrophage cell line. We found that the addition of LCA significantly suppressed TNF-α production via LPS-activated MDMs, which was consistent with responses by MH588 cells (Figure 3). In addition, LCA treatment also augmented the production of IL-10 from activated MDM, again in agreement with the results from the MH588 experiments (Figure 2). Alternatively, we found that CA suppressed TNF-α secretion via LPS-activated MDMs, suggesting that CA may have immune suppressive properties in dogs.

### 3.3. Impact of BA Exposure on TGR5 Expression

Bile acids signal to intestinal and immune cells via several receptors, particularly the receptor TGR5 [1]. Therefore, we assessed whether MH588 cells expressed TGR5 and whether exposure to BAs modulated the expression of TGR5. We found that TGR5 was highly expressed on canine MH588 cells (Figure 4). However, exposure to CA or LCA did not alter TGR5 expression, thus indicating that altered receptor expression would be unlikely to account for any observed differences between primary and secondary BAs and their effects on macrophage immune responses (Appendix A).

### 3.4. Macrophage Transcriptomic Responses to BAs

To interrogate the impact of CA and LCA more fully on canine macrophage function following LPS activation, we performed RNA sequencing (RNAseq) on MH588 cells that were stimulated for 24 h with LPS (100 ng/mL) alone or together with either CA or LCA at 25 μM.

We observed that treatment with CA prior to LPS activation significantly altered the macrophage transcriptomic response to this stimulus and resulted in the greatest number of differentially expressed genes (DEGs) compared to cells treated with LPS alone or with LPS followed by LCA (Figure 5A–C). LPS activation of the MH588 cells triggered activation of many genes associated with immune response pathways, many of which we have described recently for canine macrophages (Figure 5D; [30]). Specifically, we found that CA treatment further upregulated expression of many genes associated with activated macrophages (so-called M1 macrophages), including C-X-C motif chemokine ligand 10 (CXCL10), matrix metallopeptidase 2 (MMP2), and MAF bZIP transcription factor F (MAFF). Treatment with CA was associated with downregulation of Ras homolog family member H (RHOH), IL18R1, prostaglandin E synthase (PTGES), and nuclear receptor subfamily 5 group A member 2 (NR5A2).

In contrast, macrophage treatment with LCA was generally inhibitory to the expression of M1-associated pro-inflammatory and chemokine genes (CXCL10, MAFF), in addition to those involved with cytokine signaling (SOCS1) and extracellular matrix remodeling (MMP2; Figure 5D). Treatment with LCA following LPS activation was associated with enhanced expression of TNF-ASF13B (role in B cell activation), P2RY6, and AXL receptor tyrosine kinase (AXL), genes that are associated with anti-inflammatory effects through distinct pathways. However, LCA treatment was associated with decreased expression of IDO1 and IDO2, which would be expected to have a pro-inflammatory effect.

We also examined and compared the impact of treatment with CA versus LCA on individual macrophage-expressed genes (Table 1, Appendix A) and biological pathways (Figure 6) in LPS-activated macrophages beyond those considered to be related to M1 macrophages. Significantly upregulated genes in CA-treated, LPS-activated macrophages compared to LCA-treated macrophages include C-C chemokine motif 8 (CCL8), interleukin-1α (IL-1A), interleukin-1β (IL-1B), and C-X-C chemokine motif 6 (CXCL6). Downregulated genes associated with CA treatment when compared to LCA-treated macrophages included Src-like adapter (SLA), Fas apoptotic inhibitory molecule 3 (FCMR), and P-selectin glycoprotein ligand 1 (SELPLG) (Table 1). Significantly upregulated genes in LCA-treated, LPS-activated macrophages included IL18R1, prostaglandin E synthase (PTGES), and NR5A2 (Appendix A).

Pathway analysis using gene set enrichment analysis (GSEA) identified key immune pathways associated with macrophage responses to CA or LCA. This analysis revealed that treatment with CA upregulated inflammatory response pathways in LPS-activated macrophages involving the aryl hydrocarbon receptor, Type 2 interferon signaling pathways, and non-genomic actions of vitamin D3 (Figure 6). Pathways enhanced via LCA relative to CA exposure included nitric oxide and cell cycle regulation (E2F, G2M) pathways. LPS—lipopolysaccharide. 

Given the unanticipated similarities in LPS-stimulated cytokine production by the MDMs treated with CA or LCA, we examined the 130 genes whose expression was shifted in similar directions by these two BAs in the MH588 macrophages (Figure 5B). Protein–protein interactions showed nodes centered around innate immune genes (IL1β, IL1a, and MMP), cell-cycle-related genes (KIF4, TOP2A, and NCAPG), as well as integrins (ITGB3, ITGAV, and ITGA4; Appendix A). In keeping with this, shared KEGG pathways included Toll-like receptor and IL-17 signaling, and GO processes included negative regulation of macrophage-derived foam cell differentiation, low-density lipoprotein receptor activity, and lipoprotein metabolic processes. Thus, both CA and LCA appear to modulate immune responses and metabolic activities, in keeping with their far-reaching signaling capacities.

## 4. Discussion

Bile acids are known to play key roles in the absorption of lipids in the GI tract, but less is known regarding their immune modulatory roles. This study is the first to our knowledge to explore the immune modulatory functions of primary and secondary BAs in dogs. To address these questions, we used both canine macrophage cell lines and primary cultures of dog MDM to examine the impact of key primary (cholic acid, CA) and secondary (lithocholic acid, LCA) bile acids on macrophage activation via LPS, to which gut macrophages are continuously exposed.

Immunomodulatory impacts of BAs have been described in cells from other species but not previously in canine macrophages. Our analysis produced mixed results, with indications of differential immune modulation by primary and secondary BAs in the MH588 cells (Figure 2), as in other species [6,46]. In contrast, MDMs exposed to CA or LCA exhibited reduced TNF-α and enhanced IL-10 release compared to LPS alone (Figure 3). We noted, however, that LCA suppressed canine macrophage inflammatory responses, including significant suppression of LPS-stimulated TNF-α production, together with significant stimulation of IL-10 secretion (Figure 2 and Figure 3). Thus, canine macrophages appeared to respond to LCA as expected based on studies with human, rabbit, and rat macrophages [3,6,7,43]. This finding indicates that secondary BAs are likely to serve an important immune modulatory role in the canine GI tract.

For the current study, in addition to cytokine analysis with conventional ELISA and measurement of cell surface expression of macrophage activation markers, we also employed RNAseq to further interrogate the full breadth of BA effects on macrophage function. The transcriptome analysis of CA-treated macrophages indicated upregulation of key inflammatory genes (e.g., CCL8 and CXCL6) and pathways (e.g., aryl hydrocarbon receptor and type II IFN signaling). There was an indication of upregulation of TNF-α-related pathways by CA compared to LCA, which was not reflected by the cytokine concentrations in the MDM experiments (Figure 6 and Figure 3a). In addition, we found that CA increased expression of some inflammatory genes but failed to show significant upregulation of key genes such as those regulating TNF-α, IL-6, and IP-10 that are associated with macrophage inflammatory responses. Pathway analysis also highlighted the role of CA in stimulating genes related to the aryl hydrocarbon receptor. This receptor has key roles in tempering intestinal inflammatory responses as well as promoting antimicrobial peptide release [47], therefore providing more evidence of nuanced immune responses induced by the primary BA. Meanwhile, some features of LCA responses often countered those stimulated by LPS (suppression of innate immune genes IL1A, IL1B, and CXCL8). However, LCA was not completely immune suppressive, as, for example, we observed upregulation of the nitric oxide pathway and upregulation of indoleamine-related genes (IDO1 and IDO2). Overall, LCA appeared to be the more important BA signaling molecule in tempering LPS stimulation in these studies, which agrees with its known higher affinity for TGR5 binding compared to other BAs [6].

These transcriptomic signatures of macrophage responses to BAs thus provided important new insights into mechanisms by which BAs modulate the function of gut immune cells, extending beyond commonly studied cytokines. The canine and human transcriptomic responses to BAs by macrophages were also compared to help assess the relative relatedness of the two species. A previous study by Wammers et al. utilized a microarray platform to evaluate the impact of taurine-conjugated LCA on human primary macrophages stimulated with LPS [9]. They found that out of 865 LPS-induced transcripts, 202 were significantly modified by taurine-conjugated LCA (TLCA), with the expression of 111 genes downregulated and the expression of 50 genes further upregulated [9]. Their study identified upregulation of matrix metallopeptidase genes (MMP10, MMP12) as well as downregulation of innate immune cell activating (IL1A, CCL4) and chemokine genes (CXCL1, CXCL9, CXCL10, CXCL11, IL18) when LPS-stimulated cells were exposed to TLCA. At the protein level, they identified suppression of TNF-α release by TLCA [9]. In concordance with the findings from human secondary BA immune assays, we also found that the secondary BA LCA curtailed LPS-stimulated pro-inflammatory and chemokine gene expression in canine macrophages. Moreover, NR5A2 (also known as LRH-1) was identified as one of the most significantly upregulated genes by LCA. This gene encodes for an orphan nuclear receptor with anti-inflammatory roles and impacts on cholesterol homeostasis and bile acid synthesis [48]. Thus, multiple pathways are likely to account for the diverse impacts of BAs. Given that BA dysmetabolism and dysbiosis have been observed with chronic inflammatory intestinal disorders in both species [14,15], further investigation is warranted to examine the role of luminal BA in these patients and in dogs with CE.

Our findings suggest that there may be a plausible link between the microbiome, the metabolome, the diet, and immune responses in the gut in dogs, mediated at least in part by primary and secondary BAs. The study by Wang et al. [13] provided evidence that achievement of clinical remission through feeding of a hydrolyzed protein diet to dogs with CE correlated with increased concentrations of secondary BAs (LCA and deoxycholic acid), while concentrations of secondary BAs did not increase during dietary therapy in non-responders [13]. This study also identified the bacteriostatic properties of LCA against potential pathobionts, *E. coli* and *Clostridium perfringens*, which are associated with CE dysbiosis [49]. Our findings bolster a potential mechanistic link between the BA and microbiome shifts with dietary intervention in this population of dogs.

Beyond immune modulation, transcriptomic analysis of the MH588 macrophages pointed to impacts on metabolic activities common to CA and LCA. Recent proteomic and metabolomic studies have identified aberrant serum lipid profiles in CE dogs compared to healthy controls [50,51]. Another group found that after a trial with a hydrolyzed protein diet, CE dogs had significant alterations in serum lipid levels [52]. Bile acids could influence disease manifestation from this direction as well.

TGR5 has been previously identified using immunohistochemistry and mRNA in situ hybridization on canine cells, including macrophages, enterocytes, and enteroendocrine cells [2]. Our results are important in that they demonstrate that regulation of TGR5 expression by BAs is unlikely to account for the differential effects of CA and LCA. However, it is possible that differential binding and signaling through TGR5 (and other BA receptors) could explain how CA and LCA exert their effects on immune cells in dogs. Lithocholic acid has been reported to be among the most potent agonists of TGR5 [6], and this might explain why canine macrophages responded more robustly to lower concentrations of LCA versus CA and CDCA (Appendix A).

### 4.1. Potential Impact of the New Findings

The studies reported here provide a mechanistic basis for understanding why the absolute concentration of secondary BAs can help determine the character of immune responses in the gut in dogs with CE. Our findings indicate that the primary BA, CA, in dogs appears to have both immune suppressive and immune stimulatory properties. Thus, the absolute concentration of the secondary BA LCA may be the more important parameter to measure to assess, for example, the impact of BA dysmetabolism on regulating GI inflammation in canine CE and other conditions. It remains to be seen below what threshold concentration the anti-inflammatory effects of LCA are lost.

Diet remains the most effective therapy for canine CE [53], and the ability of diet to alter microbial populations [54] and shifts in luminal BA populations may explain in part the impact of specific GI disease-targeted dietary interventions [13]. Antibiotics are another widely implemented treatment modality for canine CE [21,55,56], and one possible mechanism for their efficacy in managing CE is through shifting the microbiome and, in turn, the BA-gut axis [55]. Counterintuitively, antibiotics commonly prescribed to dogs with diarrhea, like tylosin and metronidazole, are associated with the same disruption in fecal BA documented in CE dogs [18,19]. This may contribute to the lack of robust remission and raise questions regarding the long-term ramifications of this form of treatment. The model we describe here provides an in vitro system to further explore the connections between gut microbes, BA concentrations and ratios, and gut immune regulation and dysregulation.

### 4.2. Study Potential Weaknesses and How to Address Them

The experimental model used in these studies necessarily markedly simplified the complex gut immune milieu and focused on only a single cell type (macrophages) rather than on the full spectrum of gut mucosal cells. Exposure to different concentrations of BAs (Appendix A), as well as combinations of BAs, could have also influenced our conclusions and are important future experiments to complete to better represent the complex milieu of the intestinal lumen. The model also did not use intestinal macrophages but instead relied on a macrophage cell line and on primary cultures of monocyte-derived macrophages. Nonetheless, the model allowed for careful dissection of immune responses to two of the most prevalent BAs in the intestinal lumen and can serve for further investigations of microbial and dietary impacts on gut immune responses. The model could potentially be improved by using macrophages isolated and purified from GI biopsies, though that system too would also be imperfect due to disruption of macrophage cell functions as well as local stromal cell connections imposed during tissue digestion. The use of intestinal organoids and the incorporation of macrophages into the multi-cell epithelial structures could also be used to model BA immune responses, including more intestinal mucosal cell populations, particularly epithelial cells. The results of the RNAseq studies, particularly for the expression of individual genes but also for confirming the presence of proteins, would need to be determined with RT-PCR, immunohistochemistry, ELISA, or targeted proteomics if possible. Nonetheless, the pathway analyses performed, which incorporate the impacts of multiple different genes, provided new insights that could not be replicated by measurement of individual gene expression or protein concentration.

## 5. Conclusions

In summary, our studies are the first to report that prevalent primary and secondary BAs, CA and LCA, in dogs elicit different innate immune responses, a finding that has significant implications for linking the gut metabolome to the gut immunome. By understanding the roles of primary and secondary BAs in modulating gut innate immune responses in dogs, we now have a model to help interpret the results of BA analysis from canine fecal samples in dogs with CE.

## Figures and Tables

**Figure 1 animals-13-03714-f001:**
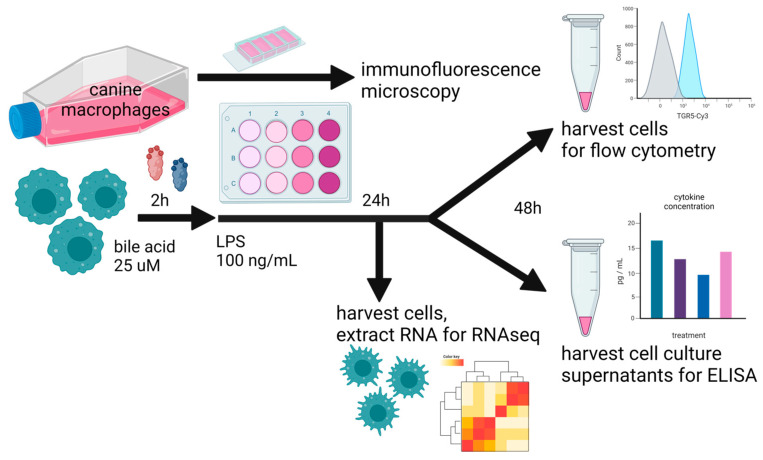
Overall study design, whereby cultured canine macrophages (MH588 cells) were plated in a 24-well plate, incubated with bile acid at 25 μM, and stimulated with LPS (lipopolysaccharide) 2 h later. After 48 h of co-culture, cells were immunostained for flow cytometry, and supernatants were harvested for cytokine analysis. For the RNAseq portion of the analysis, cells were harvested for RNA extraction after 24 h of co-culture.

**Figure 2 animals-13-03714-f002:**
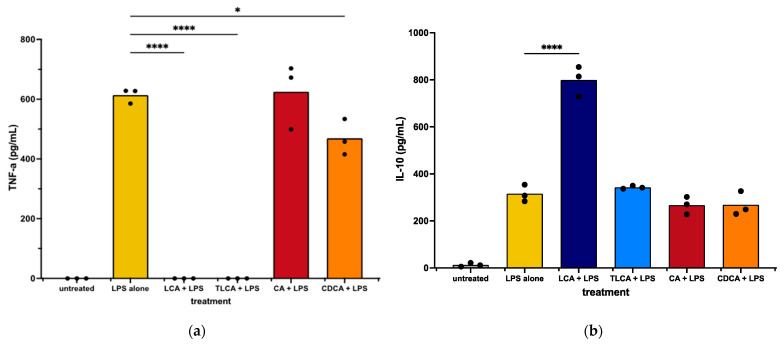
TNF-α (**a**) and IL-10 (**b**) concentrations in supernatants from canine macrophages (MH588) treated with bile acid at 25 μM, incubated for 2 h, then stimulated with LPS (lipopolysaccharide) at 100 ng/mL. Values were compared with one-way ANOVA with the Holm–Sidak multiple comparison test. Horizontal bars with asterisks indicate significantly different mean values compared to control treatment of LPS alone; * *p* < 0.05 and **** *p* < 0.0001.

**Figure 3 animals-13-03714-f003:**
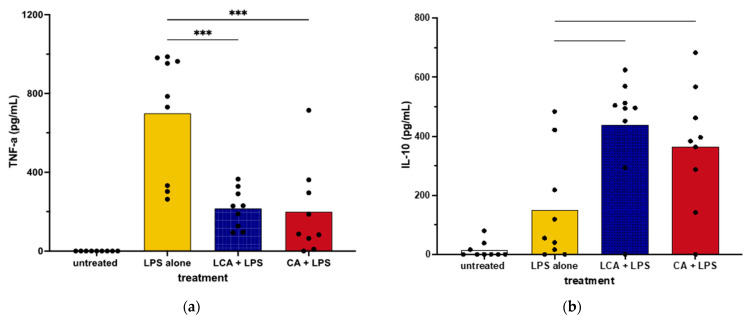
TNF-α (**a**) and IL-10 (**b**) concentrations in supernatants from monocyte-derived macrophages from healthy dogs treated with bile acid at 25 mM, incubated for 2 h, then stimulated with LPS (lipopolysaccharide) at 100 ng/mL. Values were compared with one-way ANOVA with the Holm–Sidak multiple comparison test. Horizontal bars with asterisks indicate significantly different mean values compared to control treatment of LPS alone; CA—cholic acid, LCA—lithocholic acid, *** *p* < 0.0001.

**Figure 4 animals-13-03714-f004:**
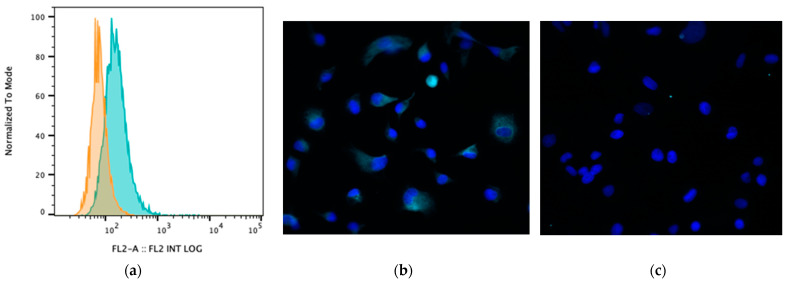
(**a**) Cell surface TGR5 expression (bile acid receptor) by MH588 cells (canine macrophage cell line) as assessed using flow cytometry. The different shadings represent different stains; orange is isotype control (rabbit IgG), and green is antibody of interest. (**b**) Fluorescence micrographs of MH588 cells at 20× magnification with unconjugated anti-TGR5 antibodies (1:50 dilution) compared to (**c**) staining with rabbit isotype control.

**Figure 5 animals-13-03714-f005:**
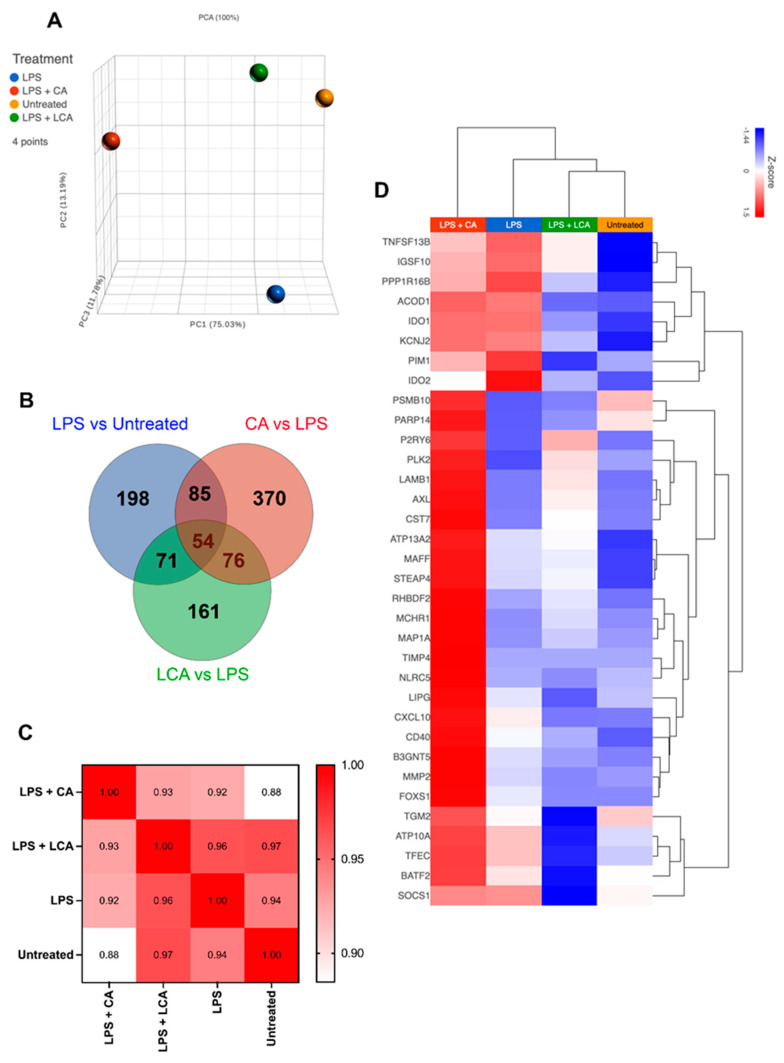
Transcriptomic analysis of LPS-activated macrophages treated with CA or LCA (LPS—lipopolysaccharide, CA—cholic acid, LCA—lithocholic acid). Macrophages were incubated with LPS and BAs (bile acids) for 24 h, then RNA was extracted for RNAseq, as described in Methods. (**A**) Principal component analysis plot showing differences in transcriptome by distance. (**B**) Venn diagram showing overlapping gene sets using differentially expressed genes defined with an arbitrarily determined *p* value of 0.05. (**C**) Pearson correlation of similarities between samples. (**D**) Heat map hierarchical clustering of M1 genes upregulated in CA compared to LCA and untreated. Gene set was extrapolated from 250 M1 genes defined in Chow et al.’s 2022 study of in vitro differentiated canine macrophages [30].

**Figure 6 animals-13-03714-f006:**
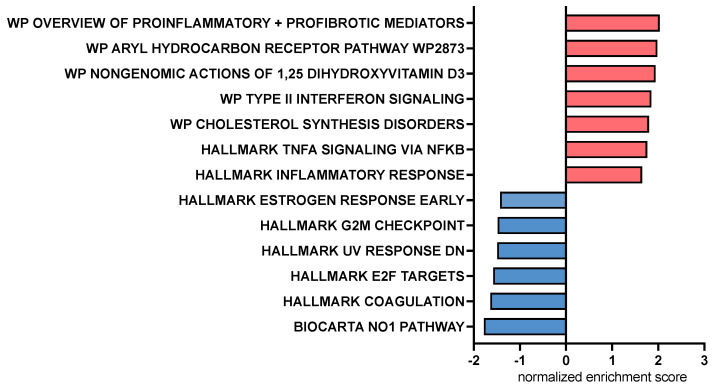
Bar chart representing pathway analysis results based on Gene Set Enrichment Analysis comparing canine macrophages exposed to CA (cholic acid, 25 μM) to those exposed to LCA (lithocholic acid, 25 μM) 2 h prior to stimulation with LPS (lipopolysaccharide, 100 ng/mL). Red bars indicate pathways upregulated by CA compared to LCA, and blue bars indicate pathways downregulated by CA compared to LCA. All displayed pathways were significantly differentially expressed with an FDR-adjusted *p* value < 0.2.

**Table 1 animals-13-03714-t001:** Summary of results comparing top 10 genes differentially regulated through exposure of canine macrophages to cholic acid (CA) compared to lithocholic acid (LCA) prior to LPS stimulation (lipolysacharide) based on gene-specific analysis. Red indicates pathways upregulated by CA compared to LCA, and blue indicates pathways downregulated by CA compared to LCA.

Gene ID	Description	Log2 (Ratio) (LPS + CA vs LPS + LCA)
CCL8	C-C motif chemokine 8	4.13
RSAD2	Radical S-adenosyl methionine domain-containing protein 2	4.10
GAP43	Neuromodulin	3.96
IL1B	Interleukin-1 beta	3.37
IL1A	Interleukin-1 alpha	3.31
ENSCAFG00000016245	Adhesion G protein-coupled receptor E2	3.10
CXCL6	C-X-C motif chemokine 6	2.75
TBX3	T-box transcription factor TBX3	2.69
SERPINB2	Plasminogen activator inhibitor 2	2.62
TIFAB	TRAF-interacting protein with FHA domain-containing protein B	2.61
SELENOM	Selenoprotein M	2.61
ENPP2	Ectonucleotide pyrophosphatase/phosphodiesterase family member 2	2.45
FBP1	Fructose-1,6-bisphosphatase 1	−6.34
PGF	Placenta growth factor	−5.26
SLA	Src-like-adapter	−5.18
FCMR	Fas apoptotic inhibitory molecule 3	−4.61
CFAP45	Cilia- and flagella-associated protein 45	−4.52
CRLF1	Cytokine receptor-like factor 1	−4.47
SLC1A7	Excitatory amino acid transporter 5	−4.40
KEL	Kell blood group glycoprotein	−4.31
SAMD11	Sterile alpha motif domain-containing protein 11	−4.26
KCNMA1	Calcium-activated potassium channel subunit alpha-1	−4.25
TMEM59L	Transmembrane protein 59-like	−3.76
SELPLG	P-selectin glycoprotein ligand 1	−3.74

## Data Availability

Raw data from the ELISA and flow cytometry experiments are available on request from the corresponding author. The RNAseq data presented in this study are openly available.

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
