# Peer review of "Modulation of In Vitro Macrophage Responses via Primary and Secondary Bile Acids in Dogs"

_animals, 2023, doi:10.3390/ani13233714_

Round 1

Reviewer 1 Report

Comments and Suggestions for Authors

The authors investigated the effect of the major primary and secondary bile acids of the gut on canine monocyte-derived macrophages by various methods. Both bile acids influence LPS-activated macrophages by increasing the secretion of immunosuppressive IL-10 and reducing TNFalpha production. Contrarily, RNA-sequencing reveals different modulated pathways. 

Overall, the manuscript is well-written, although some concerns need to be addressed:

- Introduction: In lines 267-268, the authors compare the unexpected immunosuppressive funtion of CA with contrary results in other species. Could you either include information in the introduction, that prepares the surprisingness of this finding, or even better, move this part to the discussion and give more insights into findings in other species?

- In the methods section, could you add the percentage of FBS, which specific ELISAS were used and which software was used for basic statistical analysis? Two minor issues: In CO2, the 2 needs to be subscripted; and the RNA was sent to Novogene. I like figure 1 for the understanding of the study design, but could you add the other methods, too, like MTS, FACS and IF?

- Results: In Supplemental Fig.1, the units are depicted in um instead of µm and the figure legend needs to be adapted regarding the applied CA concentrations. Additionally, I do not understand the color scheme. Why is only the 100 µM bar of LCA and CA colored, while all CDCA bars are all colored differently?

Lines 242-245: Possibly the order of sentences has been changed. As it is, the passage is unclear.

Lines 267-268: The comparison to other species is an interpretation, which at this point is not understandable for the reader, as already stated for the introduction part.

Could you please explain or correct FL2 on the y-axis of Supplemental Figure 2?

As both BA seem to act similarly regarding TNFa and IL-10 secretion, could you include information about the similarly expressed genes in the comparisons LCA vs. LPS and CA vs. LPS? Are there identifyable pathways, that indicate similar ways of action? This would also apply to the abstract and Figure 6.

In Table 1, the footer from the template needs to be removed.

How were the genes for Supplemental Figure 3 selected?

Discussion:

Lines 362 and 363: The authors state, that there is no indication of upregulation of TNFa-related pathways. Figure 6 shows upregulation of "Hallmark TNFA signaling via NFKB", could you comment on that, or explain why this pathway is not relevant for your conclusion?

The conclusion, that the ratio of BA is irrelevant, is not completely clear for me. Could you explain it in more details? In my opinion, incubation of monocyte-derived macrophages with different ratios of BA, possibly in different concentration ranges, could prove this statement. Especially since the dose-titrations experiments were performed with the cell line, which gives different results compared with those obtained with primary MDM. Additionally, the results in MDM have a high variation, despite the identified significant differences.

Two further minor issues: In line 442, the abbreviation CIE is used, probably CE was meant. In line 460, the 2 needs to be spelled out.

As TGR5 is expected to be the only receptor influencing macrophages, with varying binding affinities to the different BA, is it possible, that lower receptor activation can either be achieved by low concentrations of LCA, or high concentrations of CA? If so, would the RNA-Seq then depict different stages of BA-incluenced macrophages?

Reviewer 2 Report

Comments and Suggestions for Authors

This paper describes the in vitro effects of several bile acids on the responses of canine macrophages. As bile acid dysmetabolism is commonly recognized in dogs with chronic enteropathies this work will be of interest to the veterinary community. The paper is well written and the M&M and results support the main conclusions made. Please see some comments and suggestions below:

1.     Title: to avoid over extrapolating the study’s finding I would changes the title to 

“Modulation of in vitro canine macrophage responses by primary and secondary bile acids”

2.     Simple summary, lines 22 -24: Suggest removing “ the absolute concentration of secondary BAs such as LCA may be the most important variable to monitor for GI health.” as this speculation and there are many facets of GI healthy other than BAs.

3.     Introduction, lines 69-72: it does not appear that any of authors of the reviewed manuscript were authors of reference 16 and so I suggest removing the “we” or otherwise rephrasing.

4.     M&M, Line 10: would it be possible to make this data validating the MH588 cell line available to reader in some form (e.g., as supplemental data) or to briefly summarize flow cytometry it here?

5.     M&M, line 126: please briefly justify the physiological relevance of this BA concentration.

6.     M&M, line 12, please provide justification for the physiological relevance of this LPS concentration.

7.     M&M, line 156 b&156: has this cytokine assay be validated for use in dogs? Provide reference if possible.

8.     M&M, line 166: add a sentence explaining what the MTT assays tests

9.     M&M, 195: a fold change of +/- 1.5 seems like quite a low threshold. Please justify it with a reference if possible.

10.  Discussion, line 368: change to canine macrophages rather than dogs 

11.  Discussion, lines 420 to 422: The reviewed study does not directly support the statement that “total luminal concentration of secondary BAs, rather than the primary to secondary BA ratio, may be the most important correlate with response to dietary interventions for CE. “The information on the Wang here does not seem to directly support this. Further explanation should be provided. Also, it seems likely that this stamen is speculative and so should be deemphasized.

12.  Discussion, lines 433-435: this statement is not fully supported by this work (in part as dogs with CE were not studied) and should be soften or removed.

13.  Discussion, lines 435 to 441: this should be softened or qualified too as not all primary or secondary BAs were studied

14.  Conclusions: Confine conclusions to BAs rather than BA in general. The importance of absolute concentrations of secondary BAs seems speculative to me and should be removed from the conclusions

15.  Reference 40: details of epub can now be added:)

Comments on the Quality of English Language

See above

Round 2

Reviewer 1 Report

Comments and Suggestions for Authors

Thank you for considering and addressing all my points. Congratulations to the nice work!